# Transdifferentiation of Human Fibroblasts into Skeletal Muscle Cells: Optimization and Assembly into Engineered Tissue Constructs through Biological Ligands

**DOI:** 10.3390/biology10060539

**Published:** 2021-06-16

**Authors:** Khaled M. A. Abdel-Raouf, Rachid Rezgui, Cesare Stefanini, Jeremy C. M. Teo, Nicolas Christoforou

**Affiliations:** 1Department of Biomedical Engineering, Khalifa University, Abu Dhabi 127788, United Arab Emirates; cesare.stefanini@ku.ac.ae; 2Department of Biology, American University in Cairo, New Cairo 11835, Egypt; 3Core Technology Platforms, New York University Abu Dhabi, Abu Dhabi 129188, United Arab Emirates; rachid.rezgui@nyu.edu; 4Healthcare Engineering Innovation Center, Khalifa University, Abu Dhabi 127788, United Arab Emirates; 5Department of Mechanical and Biomedical Engineering, New York University Abu Dhabi, Abu Dhabi 129188, United Arab Emirates; jeremy.teo@nyu.edu; 6Pfizer Inc., Rare Disease Research Unit, 610 Main Street, Cambridge, MA 02139, USA

**Keywords:** transdifferentiation, direct reprogramming, 3D engineered human skeletal muscle, skeletal muscle differentiation, biological ligands

## Abstract

**Simple Summary:**

Engineered human skeletal muscle tissue is a platform tool that can help scientists and physicians better understand human physiology, pharmacology, and disease modeling. Over the past few years this area of research has been actively being pursued by many labs worldwide. Significant challenges remain, including accessing an adequate cell source, and achieving proper physiological-like architecture of the engineered tissue. To address cell resourcing we aimed at further optimizing a process called transdifferentiation which involves the direct conversion of fibroblasts into skeletal muscle cells. The opportunity here is that fibroblasts are readily available and can be expanded sufficiently to meet the needs of a tissue engineering approach. Additionally, we aimed to demonstrate the applicability of transdifferentiation in assembling tissue engineered skeletal muscle. We implemented a screening process of protein ligands in an effort to refine transdifferentiation, and identified that most proteins resulted in a deficit in transdifferentiation efficiency, although one resulted in robust expansion of cultured cells. We were also successful in assembling engineered constructs consisting of transdifferentiated cells. Future directives involve demonstrating that the engineered tissues are capable of contractile and functional activity, and pursuit of optimizing factors such as electrical and chemical exposure, towards achieving physiological parameters observed in human muscle.

**Abstract:**

The development of robust skeletal muscle models has been challenging due to the partial recapitulation of human physiology and architecture. Reliable and innovative 3D skeletal muscle models recently described offer an alternative that more accurately captures the in vivo environment but require an abundant cell source. Direct reprogramming or transdifferentiation has been considered as an alternative. Recent reports have provided evidence for significant improvements in the efficiency of derivation of human skeletal myotubes from human fibroblasts. Herein we aimed at improving the transdifferentiation process of human fibroblasts (tHFs), in addition to the differentiation of murine skeletal myoblasts (C2C12), and the differentiation of primary human skeletal myoblasts (HSkM). Differentiating or transdifferentiating cells were exposed to single or combinations of biological ligands, including Follistatin, GDF8, FGF2, GDF11, GDF15, hGH, TMSB4X, BMP4, BMP7, IL6, and TNF-α. These were selected for their critical roles in myogenesis and regeneration. C2C12 and tHFs displayed significant differentiation deficits when exposed to FGF2, BMP4, BMP7, and TNF-α, while proliferation was significantly enhanced by FGF2. When exposed to combinations of ligands, we observed consistent deficit differentiation when TNF-α was included. Finally, our direct reprogramming technique allowed for the assembly of elongated, cross-striated, and aligned tHFs within tissue-engineered 3D skeletal muscle constructs. In conclusion, we describe an efficient system to transdifferentiate human fibroblasts into myogenic cells and a platform for the generation of tissue-engineered constructs. Future directions will involve the evaluation of the functional characteristics of these engineered tissues.

## 1. Introduction

Skeletal muscle allows for the performance of essential functions such as respiration, producing locomotion, and maintaining body structure [1]. Skeletal muscle cells regenerate when subject to microtears during daily activity and this process is accentuated during exercise, to form stronger and larger tissue [2,3]. This process can be hampered by the prevalence of chronic diseases such as diabetes, genetic disorders such as muscular dystrophies, or trauma resulting in volumetric muscle loss [4,5,6]. Various in vitro and in vivo models of skeletal muscle have been developed and used to determine possible genetic or pharmacological applications tackling myopathies [7,8]. However, there are challenges in using these models: most importantly, they have yet to accurately mimic human muscular physiology and architecture [9]. Thus, the need for innovative 3D engineered skeletal muscle models persists, aiming to emulate functional contractions under various electrical, chemical, and mechanical stimuli, while also maintaining a robust microenvironment, enabling cellular communication, and providing an exchange of nutrients. Adult endogenous skeletal muscle basal lamina consists of collagen (predominantly types I and IV), that provides structural support and a medium for mechanical transduction, while also hosting laminins (particularly laminin 211) which promote cellular adhesion, alignment, and myotube formation, in addition to the recruitment of fibronectin, for the stimulation and enhancement of myogenic regeneration and proliferation [10,11,12,13]. However, recently developed reliable biological scaffolds of naturally occurring hydrogels, consisting of combinations of Matrigel^®^ and fibrin, have been thoroughly examined for skeletal muscle cellular adhesion and integration to form robust, aligned, and responsive in vitro 3D tissues with innate mimicry [14,15,16,17]. Matrigel specifically constitutes many of the vital basal lamina proteins forming in native adult skeletal muscle, such collagen IV, laminins, and sulfate proteoglycans, in addition to abundantly prominent growth factors such as Transforming Growth Factor β (TGF-β) [18,19].

The other primary component required in 3D tissue generation is a renewable cell source, with the capacity to organize into a tissue that resembles native skeletal muscle. Primary human skeletal myoblasts have been successfully used in this context, although they are characterized by limited capacity to proliferate beyond few population doublings, making their large-scale implementation challenging [20,21]. Substitute stem cell approaches have been pursued, aiming at identifying a long-term plastic cell source. Induced pluripotent stem cells (iPSCs) and mesenchymal stem cells have been differentiated into skeletal muscle cells [22,23,24]. However, without the maintenance of tightly controlled culture conditions, stem cell approaches have a susceptibility to form undesired cell types. Successful employment of stem-cell-derived skeletal muscle cells in 3D tissue engineering has been recently described [25,26,27,28,29]. A viable alternative that has gained pace with advancement in gene delivery techniques is direct reprogramming or transdifferentiation. This potentially allows for high-throughput production of adult terminally differentiated cells via the transient overexpression of targeted myogenic genes; thereby bypassing the requirement for dedifferentiation of somatic cells into iPSCs and subsequent differentiation into the desired cell type [30,31,32]. Circumventing the dedifferentiation process, the reversion of adult cells to a more nascent state of the same cell lineage, where iPSCs are derived from adult fibroblasts, via overexpression of Yamanaka factors, can be achieved via transdifferentiation of adult fibroblasts transformed directly into another adult cell lineage [33,34,35]. Efficient transdifferentiation has been achieved with neural and cardiac cells [36,37,38,39], and described in skeletal muscle, with limited throughput in the past [40,41,42,43,44]. Scientists have lately been able to transdifferentiate mouse fibroblasts into skeletal muscle cells and also assemble a 3D mouse tissue model, although a human-derived model has yet to be implemented [45].

The significance of skeletal muscle regulatory molecules is highlighted by their impact on myogenesis, control of glucose uptake, induction of regeneration, and the propagation of de novo progenitor cells [1,3,46]. Vital TGF-β proteins, the primary pathway that maintains myogenesis, such as Follistatin, have a profound ability to cause hypertrophy and muscle expansion; TNF-α can have a deleterious impact on myotube formation resulting in muscle cell necrosis, while GDF11 has yielded conflicting findings based on multiple studies emphasizing diverging results [47,48,49,50,51]. Here, we describe the transdifferentiation of human fibroblasts (tHFs), utilizing induced MYOD1 expression, into skeletal myotubes, while also differentiating C2C12 mouse and Human Skeletal Myoblasts (HSkM) in parallel experiments [52]. We evaluated the effect of a range of biological ligands (Follistatin, Myostatin (GDF8), basic Fibroblast Growth Factor (FGF2), Growth Differentiation factor 11 (GDF11), Growth Differentiation factor 15 (GDF15), human Growth Hormone (hGH), Thymosin β (TMSB4X), Bone Morphogenetic Protein 4 (BMP4), Bone Morphogenetic Protein 7 (BMP7), Interleukin 6 (IL6), and Tumor Necrosis Factor Alpha (TNF-α)), with the goal of determining their impact on skeletal muscle differentiation. We then incorporated combinations of ligands that induced a significant effect when used alone. Finally, we assembled skeletal muscle tissues derived from transdifferentiated human fibroblasts and mouse skeletal myoblasts, forming 3D compact, aligned, and multinucleated muscle tissues. 

## 2. Materials and Methods

### 2.1. Cell Culture

C2C12 mouse skeletal myoblasts (ATCC, CRL-1772), primary human skeletal myoblasts (HSkM, Gibco ThermoFisher A12555, Waltham, MA, USA), human foreskin fibroblasts (HFFs, ATCC SCRC-1041), and normal human fibroblasts (NHDFs, ATCC PCS-201-012), were propagated according to the manufacturer’s recommendations. Each cell type was cultured in fresh Dulbecco’s Modified Eagle Media (DMEM) high glucose (Gibco ThermoFisher, 11965092) supplemented with 10% Fetal Bovine Serum (FBS, Gibco ThermoFisher, 26140079), 1% L-Glutamine (Sigma, St. Louis, MI, USA, G7513), 1% Sodium Pyruvate (Sigma, S8636), and 1% Non-essential Amino Acids (Sigma, M7145) along with 0.05% Gentamicin (Gibco, 15750060), labeled growth media (GM). HSkM media was complemented with an additional 10 ng/mL recombinant human IGF-1 (R&D Systems, 291-G1-200, Minneapolis, MI, USA). Cells cultured in T75 Falcon™ (ThermoFisher) flasks, were incubated in discrete sterilized cell incubators at 37 °C, 5% CO_2_ concentration. C2C12s expanded rapidly and were cultured for two days, while HSkM and human fibroblasts (NHDFs and HFFs) were expanded over a one-week period. Cells were monitored daily with a light Axio Vert.A1 (Zeiss) microscope to observe culture growth. 

At 70–80% confluency, cells were transferred to a UV-disinfected laminar flow hood, washed once with 1x Dulbecco’s phosphate-buffered saline (DPBS, Gibco ThermoFisher, 14190250) containing no calcium or magnesium ions, and enzymatically dissociated with Trypsin-EDTA 0.25% (Gibco ThermoFisher, 25200056), then counted on a hemocytometer (Marien Field) in preparation for the respective differentiation of both C2C12s and HSkM, in addition to the transdifferentiation of human fibroblasts. 

### 2.2. Plasmid Construction and Lentivirus Preparation

A fully sequenced human MYOD1 cDNA plasmid clone (Open Biosystems, Clone ID: LIFESEQ3292930, Cat. #: IHS1380-97431575, Huntsville, AL, USA) was subcloned into the backbone of a FU-tet-on-ORF plasmid (Addgene, 19778, gift from K. Hochedlinger) [53]. To produce viral fragments for transduction, a second-generation lentiviral production system was established, combining the gene of interest (FU-tet-on-MYOD1), with a packaging vector (psPAX2, Addgene 12660), and an envelope vector (pMD2.G, Addgene 12559) hosted in HEK293T cells (ATCC, CRL-3216) [52,54]. Cells were initially expanded in GM, in a T75 Falcon™ culture flask, until 80–90% confluence. Transfection was induced in Opti-MEM (Gibco ThermoFisher, 31985070) media supplemented with 0.4% Lipofectamine 2000 (ThermoFisher, 11668019), along with three plasmids 12 µg of either FUW.M2rtTA or FU-tet-on-MYOD1, 7.7 µg psPAX2, and 4.3 µg pMD2.G. Opti-MEM was replaced 6–8 h following induction, with fresh GM, where media was collected daily for three subsequent days [52]. Accumulated viral particles were concentrated by transferring media viral supernatant into Amicon Ultra-15 centrifugal tubes (Millipore), with 100 kDa filter membranes, and centrifuging at 4500 rounds per minute (RPM), 4 °C for 30 min. This process was repeated three to four times until viral concentrate volume depressed below 500 µL. Concentrated lentivirus extract was aliquoted and stored for individual use.

### 2.3. Viral Transduction

Human fibroblasts were dissociated and reseeded onto 6 wells at 1 × 10^5^ cells/well, where lentiviral transduction was initiated by supplementation of GM with concentrated virus fragments of FUW.M2rtTA and FU-tet-on-MYOD1, at a dilution factor of 1:400, and 8 µg/mL Sequabrene (or Hexadimethrine bromide, Sigma H9268) the viral carrier [52]. Medium was exchanged the following day, and cells were ultimately re-plated prior to induction of transdifferentiation.

### 2.4. Ligand Screening and Combination Experiments

Cells (C2C12s, HSkM, and HFs) were dissociated, counted, and re-suspended in 96 well plates with GM, then differentiated or transdifferentiated for one week before exposure to biological ligands for an additional week. Briefly, cells were washed with DPBS and incubated in trypsin-EDTA, at 37 °C until cell detachment. Medium was added to dissociated cells, centrifuged, and replaced with fresh media. Cells were counted and plated in 96 well plates (ThermoFisher) with GM at a density of 20,000 cells/well for both C2C12s and transduced HFs. HSkMs were seeded at 30,000 cells/well. Medium was replaced the following day with differentiation medium (DM) for C2C12s and HSKMs, DM consisted of DMEM high glucose, 2% FBS, 1% L-Glutamine, 1% Non-Essential Amino Acids, and 0.05% Gentamicin, in addition to 10 µL/mL Insulin (Sigma, I6634) added to HSkM culture, and replenished every other day [20,55,56]. Transdifferentiation of HFs was induced with the activation of gene expression of the tet-on system with the introduction of 2 µg/mL Doxycycline hyclate (Dox, Sigma D9891) in GM sustained for one week. SB431542 (SB, 5 µM, TOCRIS, 1614) was also added to further enhance transdifferentiation towards the myogenic cell lineage [52]. Optimization of differentiation protocol of C2C12s and HSkMs was conducted via modification of seeding density (10,000, 15,000, or 20,000 cells per well of 96 well plates), the adjustment of media formulation of DM, FBS, and Horse Serum (Gibco ThermoFisher, 26050088) supplementation was tested at either 5%, 2%, or 0%, in addition to the assessment of the length of exposure to DM from 7–14 days (data not shown).

After 7 days of differentiation, cells were exposed to biological ligands previously identified as regulators of skeletal muscle development, at two different low concentrations (1 ng/mL and 10 ng/mL) for an additional 7 days and analyzed to detect discrepancies in differentiation. Proteins utilized included recombinant human Follistatin (Fs, 669-FO-025), recombinant human Myostatin (GDF8, 788-G8-010) or Growth Differentiation factor (GDF8), recombinant human basic Fibroblast Growth Factor 2 (FGF2, 233-FB-025), recombinant human GDF11 (1958-GD-010), recombinant human GDF15 (957-GD-025/CF), recombinant human Bone Morphogenetic Protein 4 (BMP4, 314-BP-010/CF), recombinant human BMP7 (354-BP-010), recombinant human Growth Hormone (hGH, 1067-GH-025), recombinant human Interleukin 6 (IL6, 206-IL-010), recombinant human Tumor Necrosis Factor Alpha (TNF-α, 210-TA-005) (All R&D Systems), and Thymosin β (TOCRIS, 3390). Ligands were reconstituted according to the manufacturer’s recommendations and aliquoted for individual use. A predetermined volume of each ligand solution was added to DM, such that the final protein concentration was 1 ng/mL and 10 ng/mL, supplemented every other day.

Following the analysis of the screening experiment results, C2C12s and tHFs were expanded and differentiated as previously described, until day 7, at which point Dox and SB431542 were removed (for tHFs) and GM (for C2C12s) were ceased and were treated with 10 ng/mL ligand combination comprising of recombinant human GDF11, TMSB4X, IL6, and TNF-α. The complete set of combinations were supplemented as single (GDF11, TMSB4X, IL6, TNF-α), double (GDF11-TMSB4X, GDF11-IL6, GDF11-TNF-α, TMSB4X-IL6, TMSB4X-TNF-α, IL6-TNF-α), triple (GDF11-TMSB4X-IL6, GDF11-TMSB4X-TNF-α, GDF11-IL6-TNF-α, TMSB4X-IL6-TNF-α), and quadruple (GDF11-TMSB4X-IL6-TNF-α) combinations, for a total of 15 unique conditions.

Following 14 days of culture, all three cell types were fixed, to prepare for immunohistochemistry. Cells were washed once with DPBS before fixation was performed in a 2% Paraformaldehyde (PFA, Electron Microscopy Science, Hatfield, PA, USA, 15713) DPBS solution soaked for 20 min. Plates were then washed and kept refrigerated in DPBS.

### 2.5. PDMS Fabrication

The fabrication of Polydimethylsiloxane (PDMS) was established from the Sylgard 184 Silicone Elastomer Kit (Dow Corning), consisted of an Elastomer Base and Curing Agent thoroughly mixed at a ratio of 10:1 (Base: Curing Agent) and poured homogeneously on Polytetrafluoroethylene (PTFE) micro-CNC-machined masters centralized in 35 mm plates (ThermoFisher). Dissolved gas particles from the mixing process were removed by placing plates in a Nalgene vacuum desiccator (Sigma-Aldrich, St. Louis, MI, USA, Z354074) for one hour. PDMS constructs were cured on a heater set at 65 °C for 4 h, then peeled from PTFE masters and stamped with a C.S. Osborne 23 mm arch punch. Oxygen plasma treatment of PDMS was conducted in a Plasma Prep III Solid State (SPI supplies) for 5 min at 100 W, 300 milliTORR. Molds were washed in absolute ethanol and soaked in Pluronic F-127(Sigma, P2443) solution (0.5% Pluronic F-127 dissolved in 1× DPBS) [57,58]. An hour later, molds were washed thrice with deionized autoclaved water and placed in 6 well plates soaked in DPBS. 

### 2.6. Construction of Engineered tHFs and C2C12 Tissue Culture

Skeletal muscle tissues were created by dissociating C2C12s and tHFs into a re-encapsulated cell solution, combined with a gel solution, and subsequently plating the mixture on pre-treated PDMS molds. Specifically, C2C12 mouse myoblasts and transdifferentiated human fibroblasts were dissociated (trypsin-EDTA 0.25%), centrifuged at 1000 RPM, 25 °C for five minutes, and counted with a hemocytometer. Cells were re-suspended in a cell solution with fresh GM at a density of 5 × 10^6^ cells/mL combined with 10 units/mL of thrombin (Sigma, T4648), while a gel solution contained GM, 5 mg/mL Matrigel (Corning, 356234) and 20 mg/mL Fibrinogen (Sigma, F8630), prepared and stored on ice (Fibrinogen was prepared in DMEM without serum). Each PDMS mold was plated with 140 µL of combined cell/gel solutions, divided into three 46 µL microwells per mold (Cell solution: 70,000 cells, 48 µL GM, and 5.6 µL of thrombin. Gel solution: 30.7 µL GM, 27.9 µL Matrigel, and 27.9 µL Fibrinogen) and left to incubate at 37 °C, 5% CO_2_ for one hour and thirty minutes for gel polymerization [59]. Tissues were kept in GM supplemented with 1.5 mg/mL 6-aminocaproic acid (ACA, Sigma A2504) for C2C12s, in addition to 2 µg/mL Dox, 5 µM SB431542, and 20 µg/mL 2-Phospho-L-ascorbic acid trisodium salt (ASA, Sigma 49752) for tHFs replaced every other day. On day 8 Dox induction and SB exposure was ceased, and GM (tHFs) along with DM (C2C12s) were supplemented with either 10 ng/mL GDF11, IL6, TNF-α (All recombinant human, R&D Systems) or TMSB4X (TOCRIS, 3390) for an additional week. Tissues were monitored daily and imaged with an Axio Vert.A1 (Zeiss) inverted microscope.

### 2.7. Cell Immunohistochemistry and Imaging

Fixed cells were first kept in a protein blocking solution for one hour to improve antibody affinity through increased permeabilization, allowing greater infiltration, and prevention of unspecific binding to random antigens. Blocking solution comprised 0.2% Triton X-100 (Sigma, T8787) and 10% FBS dissolved in 1x DPBS. All antibody dilutions were performed in block solution while all washes were completed with DPBS. Cytoskeletal Primary antibodies, Sarcomeric alpha-Actinin (anti-ACTN 2, Abcam, ab137346), and anti-MF20 (myosin heavy chain, DSHB) were prepared and aliquoted for individual use for screening experiment C2C12s, HSkM, and tHFs subjects. Anti-ACTN 2 and anti-MF20 antibodies were diluted at varying factors, anti-ACTN 2 at 1:250, anti-MF20 at 1:50, and delivered to C2C12s, HSKMs, and tHFs. Three replicate wells of each ligand condition were stained with both anti-MF20 and anti-ACTN 2. After an hour of incubation at room temperature, plates were washed thrice, to ensure the removal of bound antibodies. Combination experiments of C2C12s and tHFs were stained with anti-MyoD1 (Abcam, ab64159) and anti-Ki67 (Abcam, 15580), prepared, and mixed at 1:250 dilutions. Secondary antibodies Donkey anti-mouse (H+L) Alexa Fluor 568 (ThermoFisher, A10037) and Donkey anti-mouse (H+L) Alexa Fluor 488 (ThermoFisher, A-21202) were prepared at 1:250 serials and added to the cells after the third wash (568 added to anti-ACTN2 stained cells, 488 added to anti-MF20, anti-MyoD, and anti-Ki67 stained cells). Secondary antibody solution was removed, and cells were washed twice, before the addition of 4′, 6-diamidino-2-phenylindole (DAPI, ThermoFisher D1306), prepared at 5 µg into 50 mL of block solution for the final incubation. Cells were then washed ten minutes later, kept in DPBS, and wrapped in foil.

Samples were imaged using an Axio Vert.A1 FL (Zeiss) fluorescent microscope and captured with an AxioCam MRm (Zeiss) camera at 2.5×, 5×, and 10× magnifications. Each well of a 96 well plate was imaged in triplicates at distinct well positions, captured to incorporate the highest density of visible anti-ACTN 2 and anti-MF20 proteins, with all images taken at consistent exposure. Each ligand condition was designated two stained wells, thus six images per condition were generated. Skeletal muscle evaluation of differentiation and proliferation capacity was determined via manual data collection of fusion index, degree of multinucleation, nuclear density, myotube diameter, and length from 5× microscopy images. Fusion index was calculated by determining the ratio of quantified nuclei lying within ACTN^+^ myotubes vs. the entire nuclear field. Multinucleation was quantified via manual count of nuclei lying within ACTN^+^ myotubes, nuclear density (including Ki67 and MYOD1 nuclear density) was quantified by an average count of entire nuclei within the image field, while myotube length and diameter were measured from ACTN^+^ myotubes that lie entirely within the image field. 

### 2.8. Tissue Immunohistochemistry and Imaging

Fixed skeletal muscle tissues were carefully extracted from PDMS molds with forceps and placed in 24 well plates for immunostaining. Immunostaining protocols for both sets of tissues (C2C12s and tHFs) were performed simultaneously. All incubations were done at room temperature on a shaker, and all washes were completed in blocking solution (0.2% Triton X-100 (Sigma, T8787) and 10% FBS dissolved in 1× DPBS). Tissues were permeabilized for one hour in blocking solution before primary antibodies anti-ACTN2 (1:250 dilution) and MF20 (1:50 dilutions) were added separately and incubated for one hour. Constructs were washed three times, the final wash was left for one hour on a shaker, before the addition of secondary antibodies Donkey anti-mouse (H+L) Alexa Fluor 568 (1:250 dilutions) and Donkey anti-mouse (H+L) Alexa Fluor 488 (1:250 dilutions) to ACTN2 and MF20 stained tissues, respectively. After half an hour of exposure to fluorescent antibody solution, plates were retrieved and washed twice. The second wash was left on a shaker for thirty minutes; DAPI was added and incubated for an additional ten minutes. The final wash was left on a shaker, and tissues were kept in DPBS at 4 °C. 

Stained skeletal muscle constructs were imaged at low magnification (2.5×, 5×, and 10×) with an inverted fluorescent microscope Axio Vert.A1 FL (Zeiss). Samples were also imaged using an upright confocal microscope DM6000 M (Leica) at both low (10×, 20×) and high (63×) magnifications. Z-stacks (at 63×) and stitched whole-tissue images were also captured. 

### 2.9. Statistical Analysis

Image data were aggregated from three biological replicates (consisting of at least *n* = 3 technical replicates each), data parameters were processed on ImageJ/Fiji, and transferred to GraphPad Prism 7.0 for tabulation and graphing. Results were presented as means and standard deviation for each condition. Statistical analysis was performed using ordinary one-way Gaussian ANOVA (analysis of variances) multiple comparison testing with post hoc Tukey’s multiple comparisons analysis. Alpha level was considered statistically significant at *p* < 0.05, denoted by (*).

## 3. Results

### 3.1. Skeletal Muscle Screening of Biological Ligands

Here, we determine the effectiveness of exposure of protein ligands following skeletal muscle differentiation and investigate the impact of specific combinations of these ligands. Skeletal muscle cells were expanded, differentiated, and exposed to biological ligands to evaluate the level of differentiation based on cellular maturation and proliferation parameters. Mouse C2C12 and human primary myoblasts (HSKMs) were expanded in high serum media and differentiated into skeletal myotubes in low serum conditions for a 7-day period. Human fibroblasts were efficiently transdifferentiated into myogenic cells over a 7-day period using a recently described optimized protocol [52]: inducible overexpression of MYOD1 with concurrent exposure to SB431542 (a potent and selective inhibitor of TGF-β type I receptor ALK5 and its derivatives ALK4 and ALK7). Differentiated or transdifferentiated cells were subsequently exposed (days 8–14) to biological ligands Follistatin, GDF8, FGF2, GDF11, GDF15, hGH, TMSB4X, BMP4, BMP7, IL6, or TNF-α, previously described as playing important roles in skeletal muscle development or regeneration [60,61,62,63,64,65,66,67,68,69,70,71,72,73,74], at two concentrations (1 ng/mL, 10 ng/mL). Depiction of myogenic association of biological ligands and targeted signaling pathways have been described (Appendix A). To assess the effect of these protein ligands on differentiated cells, we assayed variables associated with differentiation or proliferation including fusion index (ratio of nuclei within myotubes), degree of multinucleation (number of nuclei incorporated per myotube), nuclear density (total number of nuclei), myotube diameter and myotube length.

#### 3.1.1. Effect of Screening C2C12s with Biological Ligands

Mouse C2C12 myoblasts were differentiated using well-established protocols (low serum medium) for 7 days. On day 8, cells were exposed to a selection of ligands at two different concentrations (1 ng/mL and 10 ng/mL) for a further 7-day period. On day 14, cells were fixed, immunostained for expression of alpha-Actinin, Sarcomeric Myosin Heavy Chain, and DAPI, and analyzed for fusion index, degree of multinucleation, cell nuclear density, myotube diameter, and length (Figure 1A,B).

Treatment with Follistatin (10 ng/mL) resulted in a fusion index increase, whereas treatment with FGF2, GDF15, hGH, BMP4, BMP7, or TNF-α (10 ng/mL) resulted in a decrease (Figure 1C); similarly, exposure to 1 ng/mL of FGF2, GDF15, BMP4, BMP7, or TNF-α resulted in a decrease in fusion index (Appendix A). FGF2, BMP4, BMP7, or TNF-α (10 ng/mL) resulted in a decrease in the degree of multinucleation (Figure 1D); although the same ligands at 1 ng/mL resulted in no observable effect (Appendix A). Additionally, we detected a large increase in nuclear density with the administration of FGF2, GDF11, or BMP4 (10 ng/mL, Figure 1E,H,I); similarly, 1 ng/mL of these ligands, in addition to TNF-α, resulted in an increase in nuclear count (Appendix A). We observed a decrease in myotube length with GDF8, FGF2, BMP4, or BMP7 while alternatively IL6 resulted in an increase (10 ng/mL, Figure 1F); furthermore, 1 ng/mL of FGF2, BMP4, or BMP7 resulted in a decrease in myotube length (Appendix A). Myotube diameter was decreased following exposure to hGH or BMP4 (10 ng/mL, Figure 1G) and TNF-α (1 ng/mL, Appendix A).

Our overall findings from the ligand screening of C2C12s indicated that FGF2, BMP4, BMP7, or TNF-α had an adverse effect on fusion index and degree of multinucleation (myogenic differentiation parameters). Myotube length or width (morphological parameters) were negatively affected to a limited degree due to TNF-α or BMP4. Nuclear density increased due to exposure to FGF2, GDF11 or BMP4. Follistatin, GDF8, GDF15, hGH, or IL6 resulted in no consistent statistically significant changes in differentiated C2C12s. 

#### 3.1.2. HSkM Response to Biological Ligand Exposure Was Limited

Primary human skeletal myoblasts were induced to differentiate in low serum medium for 7 days. Treatment with biological ligands (1 ng/mL and 10 ng/mL) was initiated on day 8 and lasted for an additional 7 days. HSkM were subsequently fixed on day 14, stained, imaged, and analyzed similarly to C2C12s (Figure 2A,B). 

We observed a decrease in fusion index in cells treated with Follistatin, GDF8, FGF2, GDF15, hGH, or TNF-α (10 ng/mL, Figure 2C); a similar decrease was observed with 1 ng/mL TNF-α (Appendix A). We observed an increase in the degree of myonucleation in cells exposed to TMSB4X or TNF-α (Figure 2D). Furthermore, we detected an increase in nuclear density in cells treated with FGF2, GDF11, or TNF-α (10 ng/mL, Figure 2E). Myotube length increased following exposure to GDF11, IL6, or TNF-α (Figure 2F), while myotube width decreased following exposure to BMP4 but increased with GDF11 application (Figure 2G). HSkMs treatment with FGF2 and TNF-α significantly increased nuclear density; however, BMP4 exposure appeared to replicate control conditions (Figure 2H–J). 

Overall, we observed a decrease in the fusion index in cells exposed to GDF8, FGF2, GDF15, and TNF-α; however, no corresponding observation was made with myonucleation, another differentiation indicator. On the other hand, proliferation, evaluated through nuclear density, increased with FGF2, GDF11, and TNF-α. Additionally, HSkM average myotube length increased following exposure to GDF11 and TNF-α.

#### 3.1.3. tHF Treatment with Biological Ligands Resulted in Decreased Differentiation with Partial Increases in Proliferation

Human dermal fibroblasts were propagated and transduced with lentiviral vectors allowing the inducible expression of MYOD1. Cells were induced to transdifferentiate into myogenic cells through the induction of MYOD1 expression (Dox) and TGFβ pathway inhibition (SB431542), for a 7-day period, as previously described [52]. On day 8, Dox and SB431542 were removed, and transdifferentiated cells were treated for an additional 7 days with Follistatin, GDF8, FGF2, GDF11, GDF15, hGH, TMSB4X, BMP4, BMP7, IL6 or TNF-α (1 ng/mL or 10 ng/mL). To assess the degree of skeletal muscle differentiation, cells were immunostained for alpha-Actinin or sarcomeric Myosin Heavy Chain, imaged, and images were analyzed to determine fusion index, degree of multinucleation, cell nuclear density, myotube diameter, and length.

Transdifferentiated fibroblasts efficiently formed multinucleated and elongated myotubes in the absence or presence of ligand treatments (Figure 3A,B). Additionally, tHFs derived skeletal muscle cells formed cross-striated and compact myotubes, whether exposed to treatment or not (Appendix A). We detected a decrease in fusion index in cells treated with FGF2 (Appendix A), BMP4, or TNF-α (10 ng/mL, Figure 3C). The degree of multinucleation also decreased with exposure to FGF2, BMP4, TNF-α, or BMP7 (Figure 3D). FGF2 exposure resulted in a large increase in nuclear cell count in tHFs at both 1 ng/mL and 10 ng/mL (Figure 3E,H and Appendix A). A decrease in myotube length was detected in cells exposed to FGF2, BMP4, TNF-α, BMP7, GDF8, GDF15, or hGH (Figure 3F), although no change was detected in myotube diameter (Figure 3G).

Overall, these data indicate that tHFs exposure to FGF2, BMP4, or TNF-α negatively affects the degree of differentiation as determined by fusion index, degree of myonucleation, or myotube length. Lesser negative effects were observed with BMP7 (degree of myonucleation and myotube length) or GDF8, GDF15, hGH (myotube length). Proliferation was only affected by FGF2, as indicated by a large increase in nuclear count. Other biological ligands, such as Follistatin, TMSB4X, and IL6 treatments, had no discernable impact on differentiation, proliferation, or morphological change.

### 3.2. Ligand Combination Contribution to Skeletal Muscle Differentiation

To further examine the effect of these ligands on cells we employed combinations of biological ligands from our previously tested group, tailored towards assessing differentiation patterns and proliferation in skeletal muscle cells. Combinations of 10 ng/mL GDF11 (G), TMSB4X (T), IL6 (I), and TNF-α (F) were selected based on the observed impact of skeletal muscle differentiation from screening experiments, in addition to their competing and contrasting roles in myogenesis and muscular remodeling [50,51,75,76,77,78,79]. C2C12s or tHFs were induced to differentiate or transdifferentiate for 7 days. Subsequently, cells were exposed to 10 ng/mL combination ligands consisting of GDF11 (G), TMSB4X (T), IL6 (I), and TNF-α (F), (15 unique ligand combinations), for an additional 7 days.

#### 3.2.1. C2C12 Ligand Combination Did Not Improve Differentiation Response

We detected a significant decrease in fusion index when C2C12 were treated with all combinations that included TNF-α (Figure 4A). A decrease in the degree of multinucleation was observed in cells exposed to TNF-α, IF, GTF, GIF, TIF or GTIF (Figure 4B). We also observed a significant decrease in nuclear density in cells exposed to TNF-α or GTIF (Figure 4C).

The proportion of MYOD1 positive nuclei increased following exposure to GTI (Figure 4D), however individual administration resulted in decreased expression of Ki67 proliferation nuclei (Figure 4E). When compared with control (Figure 4F,I), multiple ligand combinations, such as GTF (Figure 4G,J) and GTIF (Figure 4H,K), displayed a decrease in fusion index, myonucleation level, morphological data, and MYOD1^+^ nuclei, while maintaining similar expression levels of nuclear Ki67 and nuclear density. We also observed an increase in myotube cell length following exposure to GDF11, TMSB4X, and the combinations GT and GI; however, a decrease was observed with addition of TNF-α, TF, IF, GTF, GIF, TIF, or GTIF (Appendix A). We also saw a decrease in myotube diameter in cells treated with TNF-α, GF, GTF, GIF, TIF, or GTIF (Appendix A). 

Our data suggest that the level of differentiation of C2C12 cells as determined by fusion index, degree of myonucleation, and myotube length and diameter were negatively affected by TNF-α and combinations with GDF11, TMSB4X, and IL6; although we observed limited to no effect on MYOD1 nuclear expression. Proliferation indicators (density of nuclei, Ki67^+^ nuclei) produced differing results, where administration of GDF11, TMSB4X, and IL6 resulted in a decrease in Ki67, and density of nuclei was negatively affected only by TNF-α or GTIF combination. Induction of GT, GI, GF, or TI appeared to have no impact on myotube differentiation or cellular proliferation, although GTI incorporation increased MYOD1 expression. 

#### 3.2.2. tHFs Ligand Combinations Were Negatively Associated with Differentiation Capacity 

All combinations consisting of TNF-α resulted in a decrease in myogenic parameters including fusion index, degree of multinucleation, nuclear density and MYOD1^+^ expressing nuclei (Figure 5A–D), differentiation and proliferation markers. Additionally, we observed that myotube length decrease from TNF-α, IF, GTF, GIF, TIF, or GTIF (Appendix A). The number of Ki67 positive nuclei increased due to exposure to TNF-α with all of GDF11 (GF), TMSB4X (TF), and IL6 (IF) (Figure 5E). Myotube diameters were not affected (Appendix A). Comparative image analysis of standard data (Figure 5F,I) suggests conditions such as IF and GTIF exemplify large scale decreases in skeletal muscle differentiation capacity and myotube cell death, while also maintaining similar or increased (in GF, TF, and IF) Ki67^+^ and MYOD1^+^ expression levels (Figure 5G,H,J,K). Overall, treatment with GDF11, TMSB4X, IL6, GT, GI, TI, and GTI had no effect on differentiation or consistent proliferation. 

### 3.3. Tissue Engineering of C2C12s and tHFs Derived Skeletal Muscle Constructs

Engineered skeletal muscle tissues were generated by encapsulating transdifferentiated human fibroblasts tHFs or mouse myoblasts C2C12s in a fibrin/Matrigel-based scaffold, which was injected in PDMS-based molds with pillar protrusions, allowing for gel contraction and generation of mechanical loading throughout the tissue (Figure 6A). C2C12s or lentivirus transduced HFs (inducible MYOD1 expression) were transferred to a cell solution, and rapidly mixed within a gel solution as previously described [59,80]. The combined solutions were plated in PDMS molds, before adding GM supplemented with Dox and SB431542 towards transdifferentiation. Cells proliferated longitudinally and formed well-defined and organized engineered tissues via the secretion and assembly of extracellular matrix providing mechanical and structural support and concurrent remodeling of the fibrin hydrogel [8,21,57]. On day 8, cells were induced to differentiate (C2C12s) in DM, while transdifferentiation of tHFs into skeletal muscle cells was ceased with Dox and SB termination; tissues were also treated with 10 ng/mL GDF11, TMSB4X, IL6, or TNF-α for 7 additional days. Tissues were fixed on day 14, immunostained and quantified for cell diameter and cell density using confocal microscopy. Low magnification stitched microscopy of composite images were aggregated for improved visualization of stained tHF and C2C12 skeletal muscle tissues (Figure 6B and Figure 7A).

tHFs and C2C12s formed an elongated compact, multinucleated, and aligned skeletal muscle cells in a 3D tissue ECM (Figure 6C and Figure 7B); although C2C12 constructs appeared to form denser tissues, particularly towards the pillar regions of the mold (Figure 6D and Figure 7C). This was validated with high magnification confocal microscopy of each tissue set (Figure 6E and Figure 7D). We observed an increase in myotube diameter in tHFs treated with 10 ng/mL IL6 and TMSB4X, despite no change in cell density within the tissues (Figure 6F,G), indicating improved differentiation capacity without affecting cellular proliferation. However, constructs generated from C2C12 myotubes did not respond to any ligand treatment, with respect to cellular diameter and proximal cell density in 3D stacking cells (Figure 7E,F). Comparative analysis between tHFs and C2C12 derived tissues, displayed an 18–35% increase in myotube diameter and a 22.5–55.5% increase in 3D cellular density in C2C12s. This indicates C2C12s formed more compact, condensed, and more desirable tissues, while tHFs yielded limited throughput, despite treatment with biological ligands. Thus, tHFs may require additional proliferation and differentiation capacity to achieve similar proliferation and differentiation levels.

## 4. Discussion

In this study, we investigated the potential to enhance skeletal muscle cell differentiation via the addition of biologically relevant protein ligands to three types of myogenic cells: (1) immortalized C2C12 mouse myoblasts, (2) primary human skeletal myoblasts HSkM, and (3) MYOD1-induced transdifferentiated human myogenic cells. These ligands were selected for their inherent effects on skeletal muscle proliferation and differentiation in the processes of myogenesis and regeneration [50,60,62,63,65,69,81,82,83,84,85,86,87]. This work builds upon the transdifferentiation protocol we previously developed [52], to focus on the manipulation of skeletal muscle differentiation, via novel exposure to biological ligands, towards the assembly of engineered skeletal muscle tissues derived from efficiently transdifferentiated fibroblasts. Our results indicate that the myogenic differentiation of C2C12 was significantly affected following exposure to FGF2, GDF15, BMP4, BMP7, or TNF-α, with the ligands Follistatin, GDF8, hGH, and IL6 not resulting in a response. Proliferation capacity was increased between 0.8 to one-fold with 1 ng/mL and 10 ng/mL treatment of GDF11 or BMP4, whereas administration of 1 ng/mL and 10 ng/mL FGF2 resulted in a large increase in proliferation (4-fold). We determined that HSkM differentiation (fusion index) was negatively affected following treatment with GDF8, FGF2, GDF15, or TNF-α. We collected data for additional differentiation or proliferation parameters including degree of myonucleation, nuclear density, myotube length, and diameter. These yielded diverging results: including an increase in the degree of myonucleation (TMSB4X), nuclear density (GDF11, FGF2), and myotube length (IL6, GDF11), whereas TNF-α resulted in an increase in all three parameters. With tHFs, we observed a decline in differentiation capacity with BMP4, FGF2, or TNF-α, in addition to a decrease in myotube length (GDF8, GDF15, and hGH) and degree of multinucleation, (BMP7). However, proliferation was significantly increased (up to 33%) when tHFs were treated with 10 ng/mL FGF2.

By comparing the effects on the three myogenic cell types, we determined that C2C12s and tHFs displayed similar differentiation and proliferation responses when exposed to FGF2, BMP4, BMP7 or TNF-α, (decreased differentiation) and FGF2 (increased proliferation); in addition the two cell types did not respond to Follistatin, GDF8, hGH, or TMSB4X. HSkM on the other hand, exhibited a significant decrease in fusion index when exposed to TNF-α, while demonstrating contrasting data, with an increase in myonucleation, myotube length, diameter, and nuclear density. Other data suggest no established trends in differentiation or proliferation from ligand exposure.

In the context of skeletal muscle development, the importance of TNF-α, an inflammatory cytokine secreted during traumatic injury, is highlighted during the remodeling process, where it is released by immune cells to induce skeletal muscle degradation and atrophy while promoting infiltration of fibroblasts, thus deleterious on proliferation and differentiation [69,88,89]. Our data support this through the observation of a decrease in formation of mature myotubes in C2C12s and tHFs; although HSkM increases in myonucleation and myotube length has been previously observed, to a limited scale, in skeletal myoblasts undergoing stress or during initiation of proliferation [90,91,92]. BMP4 and BMP7 have been associated with skeletal muscle regulation at varying stages of myogenic and osteogenic development [66,67]. C2C12 mouse myoblasts exposed to BMP4 significantly increased proliferation rate; additionally, localized trauma instigated BMP4 expression in the surrounding vicinity of muscle injury [93]. BMP7, however, promoted differentiation of myoblasts into the osteogenic lineage and drastically decreased myogenic specific gene expression of MYOD, MYOG, and Myf5 [67,70]. Although in our study the deleterious impact of BMP4 and BMP7 on myogenic differentiation was detected in C2C12s and tHFs, the enhanced proliferation response from BMP4 was only seen in C2C12. Previous described skeletal muscle cells exposed to FGF2 did not display the ability to terminally differentiate, as the protein inhibits the key TGF-β pathway [94,95], while also drastically increasing proliferation of myoblasts [71,79,94]. We also confirmed this in our study as C2C12, tHFs, and HSkM cells displayed up to four-fold increase in nuclear density, while also exhibiting decreases in fusion index, myonucleation, and myotube length.

We proceeded to explore a combination of biological ligands GDF11, TMSB4X, IL6, and TNF-α in a total of 15 combinations of treatment to C2C12 and tHFs. These ligands were selected to counter skeletal muscle cell wasting from TNF-α, via the introduction of myokines and proteins that aid remodeling and attenuate inflammation [47,65,72,77]. TMSB4X represents such a molecule, where multiple studies have suggested its contribution towards de novo vascularization in injured tissue while also attenuating the invasive volume of neutrophils and promoting proliferation through the differentiation of native satellite and progenitor cells [76,82,96]. In a similar capacity, IL6 acts as a regulator to inflammatory effects, has been critically found to contribute to skeletal muscle growth during exercise, acting as a myokine, where it is continuously secreted via remodeling myoblasts, culminating in a 100-fold normal blood-borne increase, and bringing about hypertrophy and an attenuation to inflammation in surrounding muscle tissue [68,97,98]. GDF11 on the other hand has been described as a contentious ligand, where studies have displayed conflicting results regarding skeletal muscle maturation. Scientists have generated a parabiotic model between old and young mice, showing elevated GDF11 levels with regenerated new muscle fibers and reducing cardiac hypertrophy [50,61,99,100]; on the other hand, recombinant GDF11 delivery to cultured C2C12 and in vivo mouse models induced muscle atrophy, arrested muscle regeneration, and cardiac hypertrophy [51,101]. Our ligand combination data of both C2C12s and tHFs suggest significant loss in myogenic differentiation, up to 40%, and proliferation from all treatments that included TNF-α, despite the introduction of anti-inflammatory and myogenic promoters, TMSB4X and IL6. However, combinations of these ligands, in addition to GDF11, displayed no significant responses.

Our findings also describe the ineffectiveness of Follistatin, GDF8, hGH, in addition to the very limited impact of GDF11, GDF15, TMSB4X, and IL6. This can be attributed to multiple factors, the most evident of which are the initiation time point of treatment, dose concentration of treatment, and duration of treatment. Cells were initially cultured to allow for proliferation to sufficient quantities, before the induction of skeletal muscle differentiation for 7 days. We initiated the exposure of 1 ng/mL and 10 ng/mL biological ligands to skeletal myoblasts C2C12s, HSkM and transdifferentiated fibroblasts one week following the induction of differentiation or transdifferentiation process, while also halting small molecules Dox and SB431542 infusion (effectively terminating further transdifferentiation). Thus, ligand exposure was conducted on already differentiated or transdifferentiated cells to improve differentiation capacity towards terminal maturation. This did not capture the effect of ligand induction at the onset of differentiation (day 0) or during the proliferation process. We anticipate a heightened response from critical myogenic ligand proteins Follistatin, GDF8, TMSB4X, and IL6 on C2C12s and tHFs if treatment was initiated at the inception (day 0) of culture differentiation/transdifferentiation, based on previous analysis [47,60,62,68,102,103]. Another important aspect to consider from our results is the dose concentration of biological ligands. We proposed testing exposure at low concentrations (1 ng/mL and 10 ng/mL) to determine a benchmark for sensitivity and response towards differentiation of C2C12s, HSkM, and tHFs. Investigation of in vitro dose-dependent responses suggested working concentrations between 20–50 ng/mL of Follistatin and GDF8 [47,81,100,104], 25 ng/mL of hGH was administered to 3T3 and 10T1/2 fibroblasts to promote growth [105,106], and 40 ng/mL of IL6 infusion improved proliferative and differential patterns in low serum C2C12s [107]. We also considered the duration of ligand exposure compared to differentiation/transdifferentiation period of 7 days, where we decided from the onset to standardize it to equate in length with the differentiation phase and to occur in immediate succession. One of the most imminent limitations we face from this work is the lack of mechanistic analysis to determine the underlying gene and protein expression of downstream targets of significantly effective ligands, such as FGF2. The upregulation of WNT/β-Catenin signaling pathway, through the administration of small molecule GSK-3β suppressor glycogen synthase kinase-3 inhibitor (CHIR99021) supplemented with FGF2 mediation, has been shown to push human PSCs towards the skeletal muscle lineage via PI3-K/AKT signaling [108,109]. Thus, the evaluation of expression levels of lymphoid enhancer factor, FOXO1, SX10, or LMX1A (all WNT signaling markers) can provide validation for differentiation potential our biological ligands [110]. Additionally, Follistatin, GDF8, GDF11, and BMP4 have displayed a significant regulatory capacity towards the TGF-β pathway, thus detection of specific downstream targets activin, TORC1, and Smad proteins will provide insight into progression of skeletal muscle differentiation and proliferation from ligand treatment [48,111,112].

Finally, we were successful in generating 3D skeletal muscle tissues from tHFs and C2C12s cells in a fibrin-based hydrogel on PDMS molds. Creating a viable in vitro human skeletal muscle model, as an alternative to animal experimentation that more closely represents human physiology, necessitates the availability of a large cell source with a large capacity to terminally differentiate and display structural organization. Previously, we demonstrated the efficient transdifferentiation of human dermal fibroblasts into human skeletal muscle cells at a significant throughput via lentivirus based MYOD1 delivery and inducible expression [52]. Thus, the engineering of 3D skeletal muscle tissues derived from tHFs was conceived, forming elongated, compact, multinucleated, and aligned cells embedded in secreted skeletal muscle ECM. However, C2C12 tissues formed more densely compacted and condensed constructs, between 18–35% increase in myotube diameter and a 22.5–55.5% increase in 3D cell density, despite tHFs treatment with 10 ng/mL TMSB4X, IL6, and TNF-α. Additionally, tissues were not verified for functional applications, such as contractile force production or calcium transduction and handling [59]. We previously demonstrated the capacity of transdifferentiated fibroblasts to express calcium ion transients, however, this characterization was not replicated in cells forming skeletal muscle tissues [52]. Furthermore, the lack of evaluation of contractile force generation, such as tetanic or twitch stimulation are a significant limitation, towards achieving a skeletal muscle tissue that can mimic human physiology. As per multiple recent studies, iPSCs have served as the ideal cell source, where cells were reprogrammed into skeletal muscle myotubes, engineered to form 3D tissues, and tested for physiological functionality [25,26]. A genetically modified iPSCs model, with persistent expression of Pax7, has been generated to derive skeletal muscle tissues from Dystrophin^-|-^ myoblasts, and other forms of dystrophic cells [25], paving the way for robust, representative 3D skeletal muscle disease models. Moreover, a lentiviral Pax7 doxycycline tet-on delivery system was used to transduce iPSCs into skeletal myotubes and subsequently construct 3D skeletal myobundles [26]. These tissues have been tested for contractile tetanus and twitch force, profiled for 2D and 3D myogenic gene expression, and evaluated for engraftment and integration in a mouse model [25,26]. Thus, this model can serve as a building block for further evaluation and provide a potential road map for future implementation. As a final note, we recommend increasing duration and concentration of biological ligands to our tissue cultures, pursuing mechanistic work to identify significant signaling transduction pathways, determining gene expression levels of skeletal myotubes as a consequence of continual ligand exposure, and exposure of tissues to electrical stimulation for a sustained period (3–7 days), evidently shown to increase myotube size, calcium signaling, force production and ultimately leads towards terminal differentiation [113,114]. These refined renewable tissues have the potential to support the regenerative capacity of individuals suffering from myogenic dystrophy and other myopathies [115,116,117], in addition to serving as a distinct in vitro model that may accurately genetically, structurally, and functionally represent human physiology, for the purposes of drug screening or genetic manipulation [118,119,120].

## 5. Conclusions

Overall, our cultured skeletal muscle myotubes, derived from either human, mouse myoblasts or, transdifferentiated fibroblasts displayed sensitivity towards only a small subset of 1 ng/mL and 10 ng/mL biological ligands, revealing a negative association towards myogenesis from TNF-α (up to 40%), and to a lesser extent FGF2, BMP4, and BMP7. Furthermore, cells responded to ligands promoting cellular proliferation and inhibiting differentiation, primarily FGF2 and BMP4, by displaying up to 50% tHFs increased nuclear count and 400% in C2C12s. However, combination experiments did not yield significant improvements towards skeletal muscle differentiation. Thus, the recommended application of positively associated ligands (Follistatin, GDF11, TMSB4X, IL6, and hGH) to improve C2C12, HSkM, and tHFs differentiation is estimated between 20 ng/mL and 50 ng/mL, based on optimal literature implementation. We were able to successfully construct skeletal muscle tissues derived from transdifferentiated fibroblasts and mouse myoblasts C2C12s, forming aligned, compact, elongated, multinucleated, and cross-striated cells, phenotypically mimicking tissue constructs. These tissues are predicted to require a larger range of protein dosage concentrations, as per our 10 ng/mL dosing system did not yield statistically significant results, suggesting a range between in vitro recommended implementation and in vivo observed responses. We also recommend investigation of functional applications of skeletal muscle tissues, including exposure to electrical stimulation, assessment of contractile force production, metabolic evaluation through glucose and insulin response, and calcium channel signaling potential.

## Figures and Tables

**Figure 1 biology-10-00539-f001:**
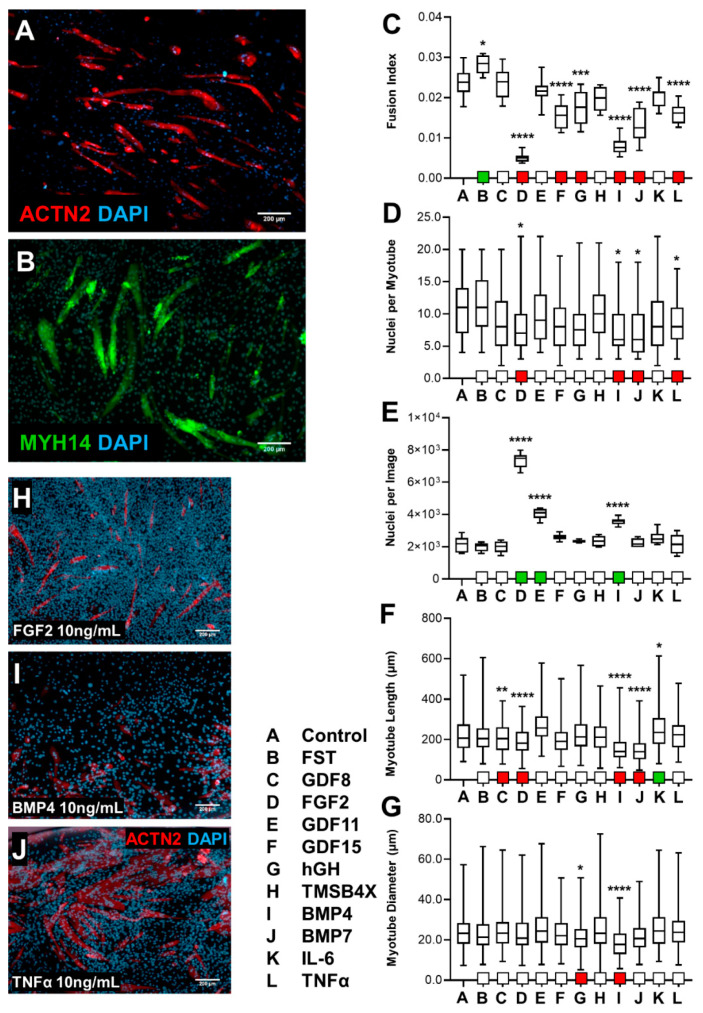
Determining the effect of biological ligand exposure on differentiation of mouse skeletal muscle cells C2C12. Biological ligand screening of C2C12 mouse skeletal muscle cells, administered at 10 ng/mL, was initiated following 7 days of differentiation. Cells were exposed for a total of 7 days, fixed, and analyzed with immunohistochemistry of anti-ACTN2 or anti-MF20 cytoskeletal stains, to evaluate the efficacy of skeletal muscle differentiation. (**A**) 5× microscopy (ACTN2) control cells. (**B**) MF20 5× immunostaining of non-treated cells. (**C**) Fusion index of C2C12s was defined as myonucleation level of myotubes as a ratio of total number of nuclei (*n* = 12, mean + SD). (**D**) Level of multinucleation of C2C12s was measured (*n* > 32, mean + SD). (**E**) Total nuclear count of both skeletal myoblasts (ACTN2−) and skeletal myotubes (ACTN2+) was quantified (*n* = 6, mean + SD). (**F**) C2C12 myotube length (µm) was measured at various ligand treatment (*n* > 135, mean + SD). (**G**) Quantification of myotube diameter (µm) from ACTN2 positive cells, per ligand condition (*n* > 160, mean + SD). (**H**) Exposure to FGF2 showed significant increase in nuclear density, and greatly decreased fusion index, multinucleation, and myotube length. (**I**) BMP4 treatment showed a reduction in differentiation (fusion index, myotube diameter and length, and nucleation) and increase in nuclear count. (**J**) Administration of TNF-α showed a decrease in fusion index and multinucleation level, without change in nuclear density and morphological features myotube length and diameter. * *p* < 0.05, ** *p* < 0.01, *** *p* < 0.001, **** *p* < 0.0001.

**Figure 2 biology-10-00539-f002:**
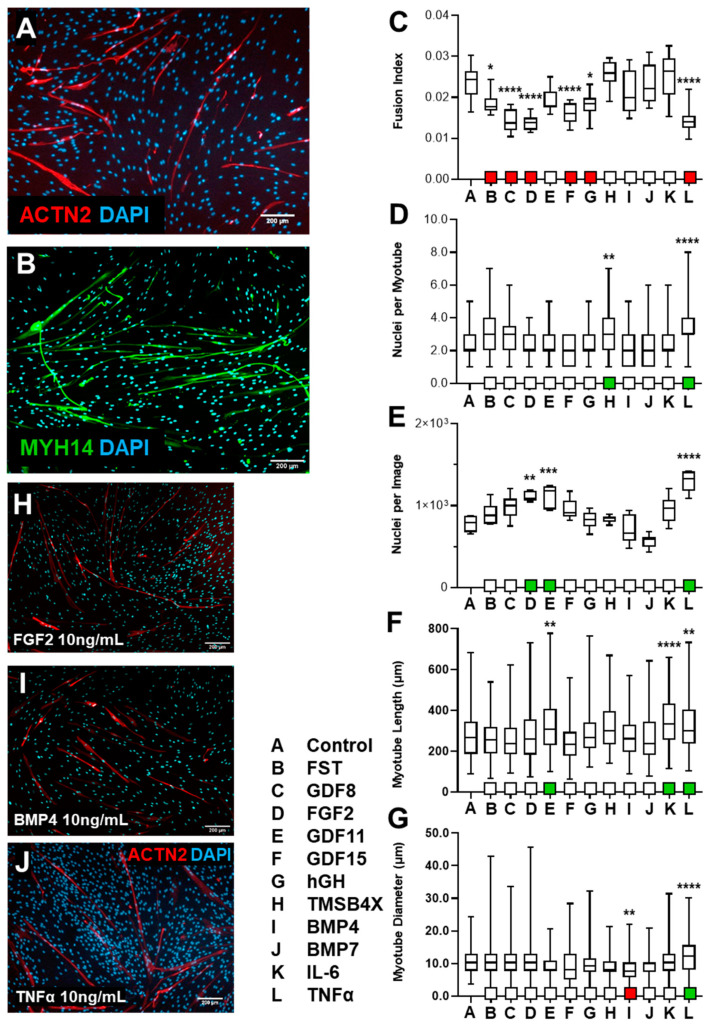
Assessment of differentiation of Human Skeletal Myoblasts (HSkM) with treatment of biological ligand. HSkM were initially differentiated for a period of 7 days before the exposure of biological ligands (10 ng/mL), for an additional 7-day period, and subsequently immunostained with ACTN2 or MF20, to determine impact on myotube differentiation. (**A**) ACTN 5× microscopy of untreated HSkM. (**B**) Cells staining with MF20 displayed elongated, multinucleated skeletal muscle cells. (**C**) Fusion index of HSkM cells was calculated by evaluating myotube nuclear density ratio with total nuclear count (*n* = 12, mean + SD). (**D**) Multinucleation levels, as single myotube nuclear count, were quantified (*n* > 45, mean + SD). (**E**) Overall nuclear count was calculated by recording the total number of nuclei per field, 5× microscopy (*n* = 6, mean + SD). (**F**) Differentiated HSkM length (µm) were derived from ACTN2+ cells (*n* > 120, mean + SD). (**G**) Myotube diameter (µm) was also measured (*n* > 150, mean + SD). (**H**) Immunofluorescence imaging of FGF2 treated cells showed increased nuclear density, although fusion index decreased. (**I**) No significant change in skeletal muscle differentiation appeared from exposure to BMP4. (**J**) TNF-α screening displayed improved multinucleation, increased nuclear count, increase in length and diameter with reduced fusion index. * *p* < 0.05, ** *p* < 0.01, *** *p* < 0.001, **** *p* < 0.0001.

**Figure 3 biology-10-00539-f003:**
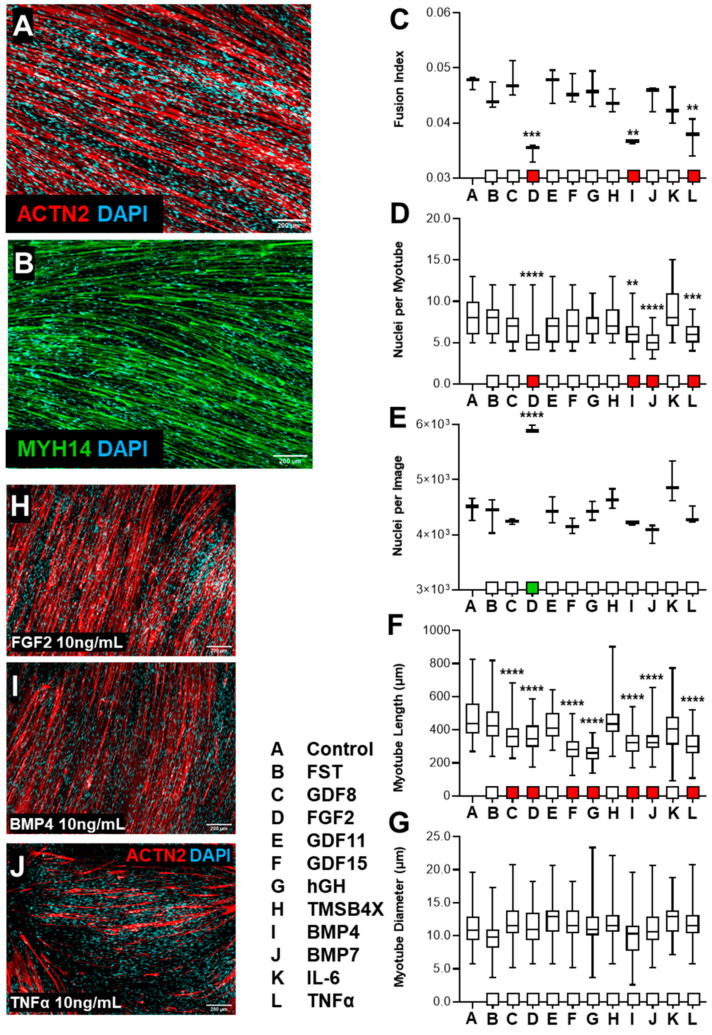
Evaluation of the differentiation of transdifferentiated Human Fibroblasts (tHFs) with the exposure of biological ligands. Human fibroblasts were induced to transdifferentiate into skeletal myoblasts via lentiviral transduction of MYOD1 gene integration, with expression promoted via doxycycline (Dox) and SB43154 (SB) incorporation. Cells were induced to transdifferentiate for 7 days with the administration of Dox and SB, and subsequently subject to ligands exposure (10 ng/mL) for an additional 7 days to determine the effect of biological ligands on skeletal muscle differentiation. (**A**) tHFs were immunostained with ACTN2 at 5× microscopy. (**B**) MF20 staining established consistent observations with ACTN2 images. (**C**) Identification of tHF myotube fusion index was derived from the ratio of myonucleation level in all myotubes vs. overall nuclear count (*n* = 3, mean + SD). (**D**) Quantification of cellular myotube nuclei was utilized to determine distribution of nuclei and thus multinucleation levels (*n* = 30, mean + SD) and (**E**) to identify nuclear density of all cells, including those of un-transdifferentiated fibroblasts (*n* = 3, mean + SD). (**F**) Myotube length (µm) and (**G**) diameter (µm) derived from ACTN2+ cells were also quantified (*n* > 60, mean + SD). (**H**) Nuclear density of cells exposed to FGF2 significantly increased, while also showing lower myotube length, fusion index, and multinucleation as compared to control. (**I**) BMP4 and (**J**) TNF-α exposure suggested limited proliferation capacity of transdifferentiated skeletal myotubes, myonucleation, and average myotube length. ** *p* < 0.01, *** *p* < 0.001, **** *p* < 0.0001.

**Figure 4 biology-10-00539-f004:**
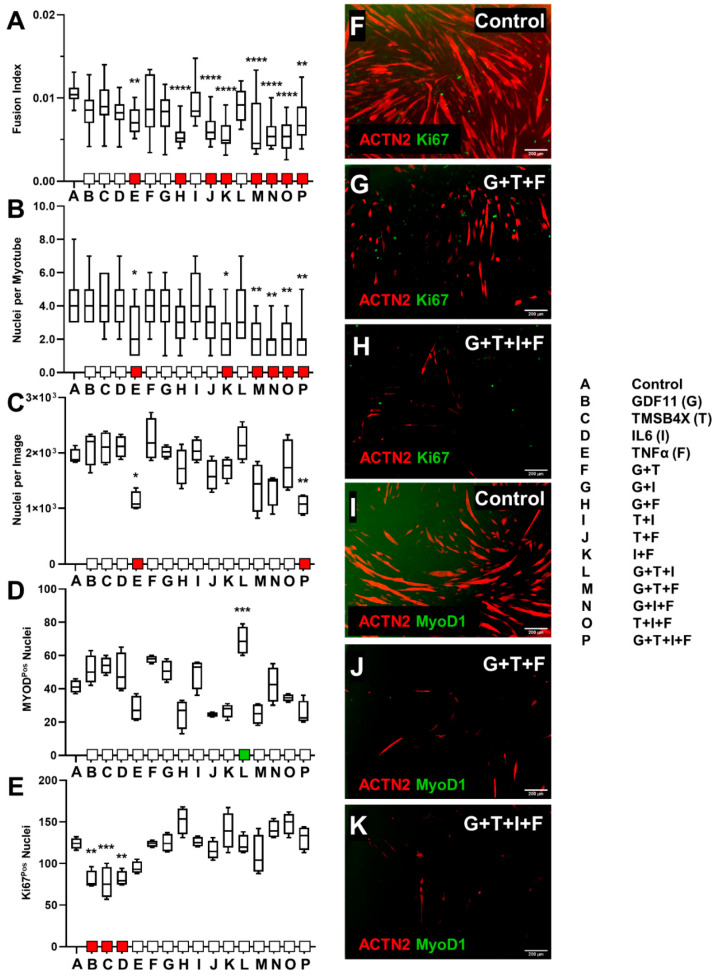
Determination of efficacy of C2C12 differentiation in conjunction with the exposure to ligand combinations. C2C12s were differentiated for 7 days and treated with combination ligands of GDF11 (G), TMSB4X (T), IL6 (I), and TNF-α (F) at 10 ng/mL for seven additional days. (**A**) Fusion index was calculated from total myotube nuclei vs. total nuclei (*n* = 16, mean + SD). (**B**) Multinucleation of C2C12 myotubes were quantified (*n* = 11, mean + SD). (**C**) Nuclear density was evaluated from nuclear count per field of 5x microscopy (*n* = 4, mean + SD). (**D**) C2C12 exposed to ligand combinations were stained to express nuclear MYOD1 (*n* = 6, mean + SD). (**E**) Cells were stained with Ki67, and where similarly quantified based on average total nuclear count (*n* = 6, mean + SD). (**F**) ACTN2 and Ki67 immunostaining of control cells. (**G**) Cells exposed to GTF showed decreases in fusion index and myonucleation levels, although no change in nuclear density and Ki67+ expression was detected. (**H**) GTIF supplementation significantly reduced skeletal muscle differentiation parameters fusion index and multinucleation, in addition to decreasing average nuclear density. (**I**) Control C2C12s expressing nuclear MYOD1. (**J**) GTF treatment greatly reduced nuclear fusion and showed limited differentiation capacity while expressing comparable levels of nuclear MYOD1. (**K**) Exposure of C2C12s to GTIF combination significantly inhibited skeletal muscle differentiation. * *p* < 0.05, ** *p* < 0.01, *** *p* < 0.001, **** *p* < 0.0001.

**Figure 5 biology-10-00539-f005:**
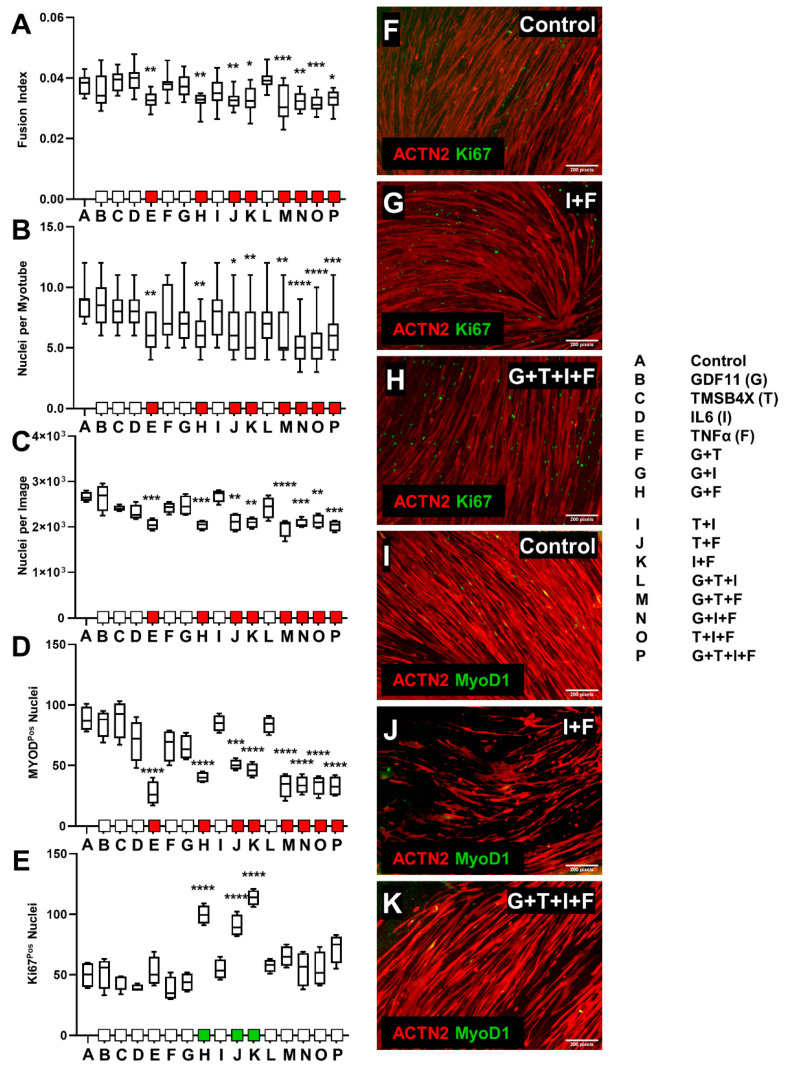
Effect of ligand combination exposure on differentiation of skeletal muscle cells derived from tHFs. Cells were transduced with MYOD1 fragments and induced to express the skeletal muscle phenotype via the induction of doxycycline and SB431542 over a 7-day period. Ligand combinations of GDF11 (G), TMSB4X (T), IL6 (I), and TNF-α (F) at 10 ng/mL were introduced for an additional week, and SB and Dox administration was discontinued. Skeletal muscle cells were fixed and stained on day 14 and characterized by various differentiation and proliferation parameters from 5× microscopy. (**A**) Fusion index of tHFs was evaluated by determining the ratio of myotube nuclei vs total nuclear count (*n* = 16, mean + SD). (**B**) Cellular multinucleation was quantified to assess tHF development of differentiation (*n* = 22, mean + SD). (**C**) Nuclear density was similarly assessed by quantifying nuclear count per field (*n* = 4, mean + SD). (**D**) Nuclear MYOD1 was quantified (*n* = 6, mean + SD). (**E**) Ki67 nuclei were also assessed with a nuclear count (*n* = 6, mean + SD). (**F**) Control tHF myotubes were immunostained with ACTN2 and Ki67. (**G**) IL6 and TNF-α combination demonstrated significant decrease in differentiation parameters fusion index, multinucleation, myotube length, and diameter (Appendix A), although Ki67+ expression had increased. (**H**) Exposure of tHFs to combined GDF11, TMSB4X, IL6, and TNF-α showed similar results, however nuclear Ki67 expression was unchanged. (**I**) Untreated tHFs with ACTN2 and MYOD1 nuclear stains. (**J**) Cells treated with IF showed a decrease in MYOD1 nuclear expression. (**K**) Additionally, GDF11, TMSB4X, and IL6 exposure yielded similar results with respect to MYOD1+. * *p* < 0.05, ** *p* < 0.01, *** *p* < 0.001, **** *p* < 0.0001.

**Figure 6 biology-10-00539-f006:**
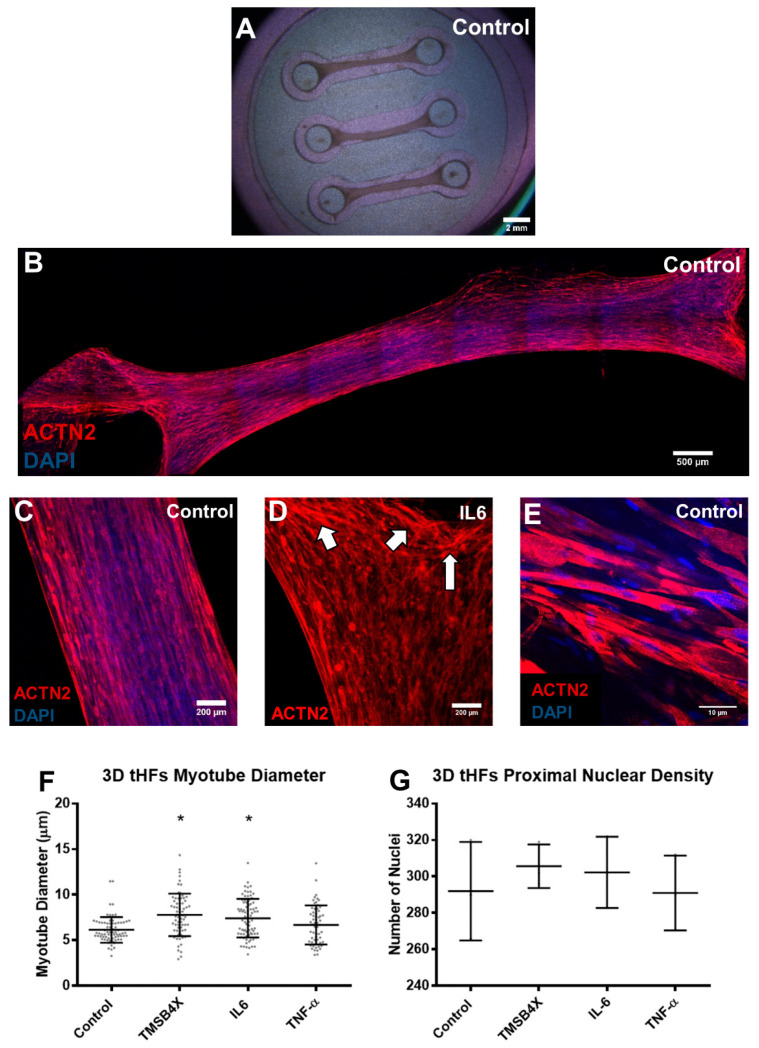
Exposure of fibrin-based engineered skeletal muscle 3D tissues derived from tHFs to biological ligands TMSB4X, IL6, and TNF-α. The transdifferentiation of HFs was initiated through the transduction of MYOD1 gene segments, encapsulated in a fibrin-based hydrogel, and plated on a PDMS mold with attachment pillars, to help develop tension within tissues. The introduction of Dox and SB instigated transdifferentiation and was sustained for a week, followed by the administration of 10 ng/mL ligands TMSB4X, IL6 or TNF-α for an additional 7 days. (**A**) Low magnification stereomicroscopy of PDMS mold, holding three microwells bearing tissues and tensile force forming around pillars, image captured on day 14 of experimentation. (**B**) Tissue-engineered tHFs were stained with Sarcomeric ACTN2, and nuclear DAPI; images were stitched at low magnification to display compact, aligned skeletal muscle cells condensed in a tissue. (**C**) Elongated, multinuclear myotube formation along the axis of the tensile force. (**D**) Cells formed condensed layers at PDMS pillars due to gradual contraction of hydrogel and accumulation of extracellular matrix. Arrows indicate condensed layers forming around pillar due to hydrogel contraction and extensive tensile force from cell-produced ECM. (**E**) High magnification skeletal myotubes show limited cross-striated cells and suggest partial compactness. (**F**) Characterization of skeletal myotube development was evaluated with diameter (µm) quantification at various cross-sectional Z-stacks (*n* > 60, mean + SD, “*” denotes significance *p* < 0.05 against control). (**G**) Total number of nuclei were identified for each condition and tallied for the entire cross-section of a proximal localization on the tissue. (*n* = 6, mean + SD, “*” denotes significance *p* < 0.05 against control).

**Figure 7 biology-10-00539-f007:**
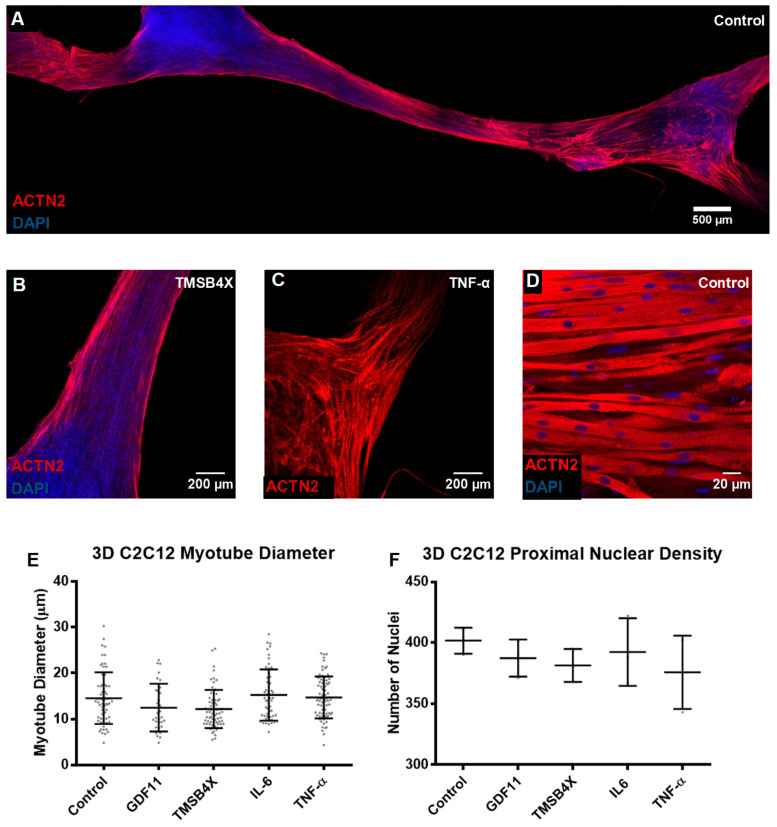
Skeletal muscle tissues were engineered from a composite fibrin/Matrigel hydrogel mixture with mouse skeletal myoblasts C2C12s, and subject to 10 ng/mL biological ligands. C2C12s were encapsulated and differentiated in a fibrin-based hybrid hydrogel over a 7-day period, 10 ng/mL biological ligands GDF11, TMSB4X, IL6 or TNF-α were administered after a week of tissue plating. (**A**) Immunohistochemical staining of C2C12 skeletal muscle constructs with ACTN2 and DAPI, demonstrated high cellular density. (**B**) Skeletal myotubes increased compactness and alignment towards central pillar regions where tensile force is maximal (**C**) Structural organization of C2C12s at pillar regions appeared disrupted due to gel contraction. (**D**) Cross-striated, multinucleated skeletal muscle form condensed tissues as demonstrated with high magnification 60× confocal microscopy. (**E**) Myotube diameter (µm) was not affected by one-week exposure to 10 ng/mL ligands. (*n* > 32, mean + SD). (**F**) Nuclear density of skeletal muscle C2C12s within tissue were not impacted with ligand administration. (*n* = 6, mean + SD).

## Data Availability

The data presented in this study are readily available on request from the corresponding authors.

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
