# Peer review of "Transdifferentiation of Human Fibroblasts into Skeletal Muscle Cells: Optimization and Assembly into Engineered Tissue Constructs through Biological Ligands"

_biology, 2021, doi:10.3390/biology10060539_

Round 1

Reviewer 1 Report

Overall comments:

The subject matter of this manuscript dealt with the transdifferentiation process of tHFs and differentiation of C2C12 cells and HSkMs by combining series of biological ligands. The roles of each component were systematically assessed by evaluating fusion index, number of nuclei, myotube length and diameter based on immunofluorescence staining of ACTN2, nuclei, and MYH14.

The present study is considered to be worth investigating and the manuscript itself seems theoretically and structurally reasonable. Moreover, the authors did well in stating what the goal of the paper is and did nicely with all the sections.

Before the next process, however, some specific concerns as commented below should be all cleared to secure its publication.

Specific concerns:

1) Line 72-89: I recommend presenting the list and role of regulatory factors in a table to help readers easily understand.

2) The substrate shown in Figure 2A does not seem to be a device that applies a load to muscle tissue through external contraction. If so, it seems that the orientation of the muscle cells has been controlled by the little tensile force generated by both supports. Can this small tensile force be a factor that promotes differentiation? If the substrate can provide sufficient tensile force, doesn’t it affect the structural integrity of the hydrogel during culture?

3) Figure 6D: Please describe in the figure caption what the white arrows depict.

Author Response

Please see the attachment below. Thank you.

Reviewer 2 Report

The manuscript by Abdel-Raouf and co-workers describes the differentiation process (on immortalized mouse myoblasts, human primary skeletal myoblasts and MYOD1-induced myoblasts from human fibroblasts) and the impact of single or multiple biological ligands on cell morphology.
The work is well-organized and rigorously designed, even if functional outcomes of the differentiated cells are lacking, as rightly highlighted by the same Authors. Some specific comments are reported below.  
- Line 297: the cited reference 43, related to an article published in 2017 by some of the Authors, describes a transdifferentiation protocol applied to human dermal fibroblasts to obtain skeletal muscle cells by MYOD1 induction. The Authors should highlight the novelty of the current work with respect to ref. 43. 
- Lines 531-532: "... and concurrent disintegration of the fibrin hydrogel..." Please describe how the gel disintegration was assessed.
- Figure 1 is shifted on the right so that it appears partially trimmed.
- Please check English language throughout the manuscript, correcting typos and grammar.

Author Response

Please find the attachment below. Thank you.

Reviewer 3 Report

This manuscript summarizes the attempt of the authors to   and to use these cells to build an in vitro construct on PDMS substrates. The described protocol efficiently enable to culture skeletal muscle myotubes, derived from either human, mouse myoblasts or, transdifferentiated fibroblasts. These appear sensitive to 1 ng/ml and 10 ng/ml biological ligands, that promote cellular proliferation and inhibiting differentiation. Skeletal muscle differentiation was not observed with combination experiments. With higher ligands doses (Follistatin, GDF11, TMSB4X, IL6 and hGH at 20 ng/ml and 50 ng/ml) cell differentiation is obtainable.

I have included comments in the revised version, in a pdf file.

There are some points to improve to expand the impact of the work:

  • the final evidences of tissue constructs are an important feature of the work and should be linked with a more detailed reference to the literature. Some groups have attempted to build and use similar constructs using different cell sources and integrated these with active platforms for testing contraction, measure calcium channel signaling potential and metabolic response to glucose and insulin response. Please include further literature and examples of platforms or tissue constructs that could benefit by the use of the proposed cells. 
  • Figure resolution: please use higher resolution figures and correct some misplacement and format. 

Author Response

(The authors gave the same response as above.)

Round 2

Reviewer 2 Report

The Authors accomplished the Reviewer requests. Even if the lack of functional and mechanistical experiments dramatically limits the soundness of the work, these issues are clearly stated in the manuscript. Thus, I recommend publication of the revised work.

Reviewer 3 Report

I appreciate the Authors improved the manuscript addressing the comments and questions I made. The manuscript is now in my opinion acceptable for publication.